



# Contrasting drought legacy effects on gross primary productivity in a mixed versus pure beech forest

Xin Yu[1], René Orth[1], Markus Reichstein[1], Michael Bahn[2], Anne Klosterhalfen[3], Alexander Knohl[3], Franziska Koebsch[3],

Mirco Migliavacca[1, 4], Martina Mund[5], Jacob A. Nelson[1], Benjamin D. Stocker[6, 7], Sophia Walther[1], and Ana Bastos[1]

[1]Department of Biogeochemical Integration, Max Planck Institute for Biogeochemistry, Jena, D-07745, Germany
[2]Department of Ecology, University of Innsbruck, Innsbruck, A-6020, Austria
[3]Bioclimatology, University of Göttingen, Göttingen, D-37077, Germany
[4]Joint Research Centre, European commission, Ispra (VA), 21027, Italy
[5]Forestry Research and Competence Centre Gotha, Gotha, D-99867, Germany
[6]Department of Environmental Systems Science, ETH, Zürich, 8092, Switzerland
[7]Swiss Federal Institute for Forest, Snow and Landscape Research WSL, Birmensdorf, 8903, Switzerland

*Correspondence to*: Xin Yu (xyu@bgc-jena.mpg.de)

**Abstract.** Droughts affect terrestrial ecosystems directly and concurrently, and can additionally induce lagged effects in subsequent seasons and years. Such legacy effects of drought on vegetation growth and state have been widely studied in tree-ring records and satellite-based vegetation greenness, while legacies on ecosystem carbon fluxes are still poorly quantified and understood. Here, we focus on two ecosystem monitoring sites in central Germany with similar climate but characterized by different species and age structures. Using eddy-covariance measurements, we detect legacies on gross
primary productivity (GPP) by calculating the difference between random-forest model estimates of potential GPP and observed GPP. Our results showed that at both sites, droughts caused significant legacy effects on GPP at seasonal and annual time scales which were partly explained by reduced leaf development. The GPP reduction due to drought legacy effects is of comparable magnitude to the concurrent drought effects, but differed between two neighbouring forests with divergent species and age structures. The methodology proposed here allows quantifying the temporal dynamics of legacy
effects at the sub-seasonal scale and separating legacy effects from model uncertainties. Application of the methodology at a larger range of sites will help quantify whether the identified lag effects are general and on which factors they may depend.

## 1 Introduction

The frequency, intensity, duration, and spatial extent of drought are expected to increase in the next decades due to anthropogenic global warming in many regions (IPCC, 2022). A great number of studies, considering both long-term
observations (Schwalm et al., 2010; Zscheischler et al., 2014) and model simulations (Reichstein et al., 2007; Sun et al., 2015) across various spatial scales, have shown that droughts concurrently impact the structure and function of terrestrial ecosystems (Assal et al., 2016; Frank et al., 2015; Lewis et al., 2011; Ma et al., 2015; Orth et al., 2020), potentially turning



ecosystems from sinks to temporary sources of carbon (Ciais et al., 2005; Reichstein et al., 2013). Therefore, understanding the impact of droughts on terrestrial ecosystems is a key research question in Earth sciences (Piao et al., 2019).

Drought impacts on terrestrial ecosystems are not limited to concurrent effects, but also include legacy effects during the following seasons and years (Anderegg et al., 2015; Frank et al., 2015; Kannenberg et al., 2020). Legacy effects at tree and/or stand scale can be caused by the higher vulnerability to drought  due to previous water depletion of the soil (Krishnan et al., 2006, Galvagno et al., 2013), reduced or delayed leaf development (Migliavacca et al., 2009; Rocha and Goulden, 2010; Kannenberg et al., 2019), drought-induced hydraulic damage of the xylem (Anderegg et al., 2013), adjustments in

carbon allocation within the trees (Huang et al., 2021), depletion of non-structural carbohydrates (Peltier et al., 2021) due to reduced carbon availability and adjustments in carbon allocation (Hartman and Trumbore, 2016), tree mortality (Allen et al., 2015), as well as reduced resistance to disturbances (e.g. insects outbreaks) due to depleted non-structural carbohydrates (Erbilgin et al., 2021). However, at the ecosystem level the impact of species and age structures on legacy effects are still less understood (Haberstroh and Werner, 2022, Wang et al., 2022).

Tree-ring records cover periods of decades to centuries and can cover multiple drought events, being therefore widely used to analyze inter-annual legacy effects of drought on tree growth (Anderegg et al., 2015; Huang et al., 2018; Kannenberg et al., 2019). Beyond the level of individual trees, satellite-based observations and model outputs, as expressed through vegetation greenness (Wolf et al., 2016; Wu et al., 2018), canopy backscatter (Saatchi et al., 2013), aboveground carbon stocks (Wigneron et al., 2020), and gross primary productivity (Schwalm et al., 2017, Bastos et al., 2020)  have also been

used to study seasonal and inter-annual legacy effects of drought. However, studies focusing on carbon fluxes, especially based on eddy-covariance measurements, are still rare (Kannenberg et al., 2020). Eddy-covariance data with hydrometeorological variables measured in parallel have the potential to quantify the timing and magnitude of legacy effects at the sub-seasonal and annual scales, and might provide insights into the mechanisms of legacy effects that might not be fully reflected in vegetation indices and tree rings.

Assessments of drought impacts on the ecosystem carbon fluxes usually focus on direct and concurrent effects (Ciais et al., 2005; Reichstein et al., 2007) without considering legacy effects. This is probably due to the challenge to attribute signals in the observations to a previous drought and hence identify them as legacy effects on ecosystem carbon fluxes (Kannenberg et al., 2020), and the inability of models to reproduce these legacy effects (Bastos et al., 2021). A number of studies consider ecosystems to have 'recovered' when the target variable such as gross primary productivity (GPP) and tree-ring width

returns to the baseline, which is usually based on pre-drought values of the target variable (Bose et al., 2020; González de Andrés et al., 2021; Zhang et al., 2021). However, this might complicate the detection of legacies since GPP recovery dynamics is affected by hydrometeorological conditions in legacy years, which can either stimulate or slow-down recovery. Here, by estimating potential GPP given hydrometeorological conditions in legacy years, we consider that 'recovery' happens when the actual GPP reaches the potential GPP under the given hydrometeorological conditions, rather than the

absolute flux.



Therefore, we aimed to develop a novel approach to quantify drought legacy effects on GPP at the sub-seasonal and annual scales. To do this, we followed a residual approach (Beringer et al., 2007) to identify legacy effects as the residuals between actual and potential GPP which is estimated by a machine-learning algorithm (specifically Random Forest regression). Furthermore, it is crucial to understand if the residuals are caused by model uncertainties or can be interpreted as legacy effects. By overlooking model uncertainties, one could misinterpret small residuals as 'legacy effects'. Here we quantified model uncertainties to provide more robust estimates of drought legacies and avoid misinterpretation of results. To test our approach, we used eddy-covariance measurements at two neighbouring sites that experienced similar climate but are characterized by different species and age structures in central Germany. We asked 1) can we detect drought legacy effects on GPP? 2) is the GPP reduction due to drought legacy effects significant compared to the magnitude of drought concurrent effects? 3) how do drought legacy effects on GPP differ at two neighbouring forests with different species and age structures?

## 2 Data

### 2.1 Study sites

The two neighboring temperate forest sites studied here, Hainich (DE-Hai, 51°04′46″N, 10°27′07″E) and Leinefelde (DE-Lnf, 51°19′42″N, 10°22′04″E), are located in central Germany, approximately 30 km from each other. These two sites share similar climate conditions, with long-term annual mean of 8 °C for 2-m air temperature and 750 mm of total annual precipitation (Tamrakar et al., 2018). Both sites were affected by the two extreme central European droughts in 2003 and 2018 which reduced gross primary productivity (Fu et al., 2020; Herbst et al., 2015).

The forest at Hainich is an old-growth, uneven aged (1-250 years) mixed forest, dominated by beech (*Fagus sylvatica*, representing approximately 64% of the tree carbon stocks). Ash (*Fraxinus excelsior*, 28%) and sycamore (*Acer pseudoplatanus*, 7%) are co-dominant tree species, and additionally there are few trees of European hornbean (*Carpinus betulus*), Norway maple (*Acer platanoides*), and other deciduous species (Knohl et al., 2003). The forest at Leinefelde can be characterized as a managed even-aged (ca. 130 years) pure beech forest (Anthoni et al., 2004).

### 2.2 Eddy-covariance and meteorological measurements

Identical eddy-covariance instrumental setups and data acquisition techniques were carried out at the two sites. The methodology of data collection and quality control followed those of Aubinet et al. (2000). The standard processing methods (Pastorello et al., 2020) adopted by the Integrated Carbon Observation System (ICOS) were used to carry out the gap-filling and the partitioning of net $CO_2$ exchange (NEE) into gross primary production (GPP) and ecosystem respiration ($R_{eco}$) using the nighttime partitioning algorithm (Reichstein et al., 2005). A detailed description of meteorological data and instrumentation can be found in previous studies (Anthoni et al., 2004; Knohl et al., 2003). We used daily meteorological data alongside carbon and water fluxes, namely GPP, latent heat flux after the energy balance correction (LE_CORR), which





was converted to evapotranspiration (ET) using the heat of vaporization, incoming shortwave radiation (SW_IN), air temperature (TA), vapor pressure deficit (VPD), soil water content at the first layer (SWC_1, 8cm), the second layer (SWC_2, 16cm), the third layer (SWC_3, 32cm), and potential incoming shortwave radiation (SW_IN_POT) for the years 100  2000-2020 at DE-Hai and 2002-2012, with a gap in 2007-2009, at DE-Lnf.

Additionally, we used daily enhanced vegetation index (EVI) data from the FluxnetEO v1.0 dataset (Walther et al., 2021) for the same years as the eddy-covariance data. EVI was derived from the MCD43A4 product of MODIS with a 500m spatial resolution and we used an average over 2x2 pixels surrounding the tower. We further estimated daily transpiration based on the Transpiration Estimation Algorithm (Nelson et al., 2018).

### 2.3 Tree ring width and net primary productivity of fruits and leaves

Annual mean tree ring width (TRW) was measured via permanent band dendrometers. The dendrometer trees represented the main species and their respective size classes at DE-Hai for the years 2003 to 2020. Because of technical constraints, damages and a natural dieback of single trees, the number of measurement trees per year varied between 54 and 95. Net primary productivity (NPP) of fruits for the years 2003 to 2020, and NPP of leaves for the years 2003 to 2016 resulted from 110  litter samplings (25-29 traps). The high fluctuation of annual fruit NPP is caused by the periodically high fruit production (masting) of beech (*Fagus sylvatica*). In mast years the proportion of beech fruits (nuts and shells) amounted to almost 92% of total fruit mass. At DE-Lnf these data are not available. A detailed description of measurement and processing methods can be found in a previous study (Mund et al., 2020).

### 3 Methodology

### 3.1 Data processing

As the first step, we filtered and processed the eddy covariance and meteorological data in the following way:

1) To ensure reliable data for our analysis we used gap-filled daily data for days for which more than 70% of measured and good quality gap-fill data were available.

2) We only used data during the growing season which was defined as the period when GPP was greater than 10% of 120  maximum of GPP as inferred from a smoothed (centered 7-days moving averages) daily average GPP across all years.

3) We calculated anomalies of all variables by subtracting the mean seasonal cycle and any significant long-term linear trend, detected by the Mann-Kendall test (Kendall, 1948), as these can obscure drought-related signals. We took the mean of each day across all considered years and then used centered 7-days moving averages to calculate the mean seasonal cycle.

4) Furthermore, a 7-days moving average smoothing was applied to the anomaly time series to filter out noise at daily time 125  scales. We expect this to increase the accuracy of our model while preserving drought legacy patterns which rather/better emerge at longer time scales.





As for TRW data, we removed for each individual tree any significant long-term linear trend detected using the Mann-Kendall test (Kendall, 1948).

## 3.2 Water availability index estimation

Soil moisture at the two study sites was measured only at the upper 30 cm and thus does not account for water availability in deeper layers (see Section 5.4). Therefore, we used a bucket model approach based on observed evapotranspiration and precipitation to estimate a vegetation water availability index, WAI (Tramontana et al., 2016), calculated as:

$$WAI_t = min(WAI_{max}, WAI_{t-1} + P_t - ET_t) \tag{1}$$

Where $WAI_t$ (mm), $P_t$ (mm), and $ET_t$ (mm) were water availability index, precipitation, and evapotranspiration at time step $t$. We set the bucket size (i.e. $WAI_{max}$) as the maximum cumulative water deficit (CWD) at each site. The estimated bucket sizes were 205mm and 191mm at DE-Hai and DE-Lnf, respectively.

Additionally, we calculated the CWD, which was estimated from cumulative differences between observed evapotranspiration and precipitation over periods where cumulative net water loss from the soil ($\Sigma$ (ET-P)) is positive.

To initialize the bucket model, we ran it 5 times through the first year before starting the actual computation across all considered years.

## 3.3 Drought and legacy years selection

Since legacy effects should result from significant impacts of droughts on ecosystems, we adopted a combined driver and impact-based approach to define droughts. Drought years were defined as those years when both low water availability and a concurrent biospheric response were found, and were evaluated as follows:

1) First, we selected the minimum of negative GPP anomalies relative to the mean seasonal cycle during the growing season (minimum $GPP_{anom}$) as a proxy to reflect the severity of drought impact on GPP in each year.

2) Then, we calculated the mean WAI anomalies relative to the mean seasonal cycle for days when minimum $GPP_{anom}$ occurred and the previous 14 days (mean $WAI_{anom\_15}$) to reflect the water availability during the development of the GPP anomaly. To identify drought-related GPP reductions, we considered only years where negative GPP anomalies were associated with dry conditions.

3) Finally, we selected the years with both the lowest minimum $GPP_{anom}$ and mean $WAI_{anom\_15}$ (Fig. S1). These were 2003 and 2018 at DE-Hai and 2003 at DE-Lnf (2018 data not available here).

In our data, we define non-legacy years as normal and drought years, while legacy years correspond to the two calendar years following a drought year. Including too few legacy years could lead to an underestimation of legacy effects, and too many legacy years would result in the lack of training data (see Section 3.4). As a trade-off, we selected a legacy period of two years and this choice was justified by the fact that GPP anomalies residuals returned to the range of model uncertainties



(i.e. 25$^{th}$-75$^{th}$ percentiles of model residuals) in 2005 (see Section 4.3) following the 2003 drought at both sites and, for 2018 at DE-Hai, data was only available up to 2020.

## 3.4 Quantification of legacy effects on GPP and transpiration

Here, we followed a residual approach (Beringer et al., 2007) to detect drought legacy effects on GPP. To do this, we fitted a random forest regression model (RF, Breiman 2001) for daily GPP anomalies using the anomalies of hydro-meteorological variables in non-legacy years as predictors. We chose RF because it has the ability to effectively learn 1) the relationship between independent and dependent variables regardless of linear or non-linear relationships; 2) the interactions between independent variables (Ryo and Rillig, 2017). The model was then used to predict GPP anomalies in the legacy years, thereby reflecting the potential GPP anomalies given the climate conditions in that year. Specifically, the approach included the following steps (Fig. 1):

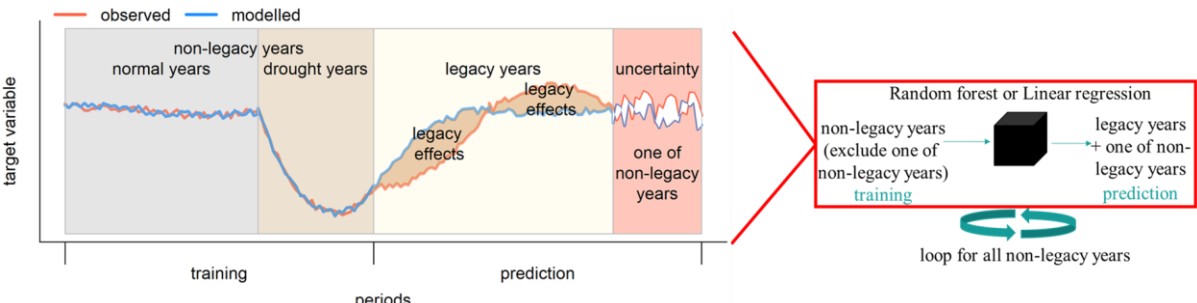

**Figure 1. Conceptual diagram of quantification of legacy effects.** A random forest (RF) model (or linear regression, represented by the black cube on the right) was used to determine the relationship between the target variable (GPP$_{anom}$ or TRW) and hydro-meteorological conditions using a training dataset excluding data in legacy years and one of non-legacy years for each loop. The legacy effects could be quantified as the residuals between observed (red line) and modelled (blue line) target variable (i.e. GPP$_{anom}$, TRW, …) in legacy years. And the residuals between observed and modelled target variable (i.e. GPP$_{anom}$, TRW, …) in all non-legacy years from all loops indicated RF model uncertainties using a leave-one-out approach (see below).

First, all daily data in non-legacy years were used as input for the RF model to determine the relationships between anomalies of GPP (GPP$_{anom}$) and anomalies of hydro-meteorological variables (SW_IN$_{anom}$, TA$_{anom}$, VPD$_{anom}$, and WAI$_{anom}$) along with absolute values of SW_IN_POT to capture seasonal variations in the response of ecosystems to hydro-meteorological conditions. These relationships represented long-term controls of climate on GPP, including drought events and near-average or wet conditions. The 'randomForest' package in R 4.0.3 was used, and the number of trees, the number of variables randomly sampled as candidates at each split, and the node size of RF were set to 400, 5, and 5, respectively. Tuning those hyperparameters did not significantly change our results.

Based on these relationships and the meteorological anomalies in legacy years, we used the trained RF model to predict the potential GPP$_{anom}$ in the absence of legacy effects and calculated the model's residuals (GPP$_{anom}$ residuals, i.e. observed minus predicted values), which should reflect legacies from the past drought: negative residuals corresponded to more negative or less positive GPP$_{anom}$ than would be expected given the meteorological conditions in that year, indicating





negative legacies of drought, while positive residuals corresponded to less negative or more positive GPP$_{anom}$, indicating beneficial legacies of drought. In order to reduce the noise at the daily scale, daily results were aggregated to the weekly scale.

To account for model uncertainties and evaluate the significance of legacy effects, we used a leave-one-out approach to

190 quantify model uncertainties. In the training phase, one of the non-legacy years was excluded from the training dataset and the trained RF model was then used to predict the GPP$_{anom}$ in that year. This was done for all non-legacy years, and the GPP$_{anom}$ residuals in non-legacy years for each leave-one-out iteration were then considered as model uncertainties. In order to reduce the noise at the daily scale, all the daily results were aggregated to the weekly scale.

The same method was used to quantify legacy effects on transpiration (Tr).

## 3.5 Quantification of legacy effects on tree growth

To detect legacy effects on tree growth, we used a multivariate-linear regression instead of RF to develop the relationship between tree growth (detrended tree ring width, TRW) due to the fewer data points available. We used the following explanatory variables: detrended annual mean WAI, detrended annual mean VPD, detrended annual mean SW_IN, and detrended annual mean TA for each species. Annual net primary productivity of fruits (fruits-NPP) particularly was added as

an additional predictor to only the model for beech since the high fluctuation of annual fruit NPP could be caused by the periodically high fruit production (masting) of beech. We considered fruits-NPP as a predictor to account for the trade-off between tree growth and reproduction in mast years, which could also cause the change in tree growth in addition to legacy effects from previous droughts (Hacket-Pain et al., 2015).

The strategy to quantify legacy effects and model uncertainties was the same as in the case of GPP. We trained the model in

non-legacy years except for each one of them iteratively and predicted potential TRW in legacy years and the year additionally excluded. The residuals between observed and potential TRW in non-legacy years and legacy years were then considered as model uncertainties and legacy effects, respectively.

## 3.6 Separation of legacy effects on GPP due to structural and physiological effects

Drought legacy effects on GPP might result from changes in canopy structure (structural effects) and photosynthesis capacity

(physiological effects) (Kannenberg et al., 2019). Combining GPP and satellite-based EVI allows separating these structural and physiological effects. To do this separation, we used two model settings: 1) RF, which was the original setting described in section 3.4, included both structural and physiological effects; 2) RF$_{EVI}$, which added EVI anomalies as an additional predictor to the original model, only include physiological effects because taking structural effects reflected by EVI anomalies into account. Therefore, physiological legacy effects on GPP were quantified as GPP$_{anom}$ residuals from RF$_{EVI}$

while structural legacies were quantified as the difference between GPP$_{anom}$ residuals from RF and RF$_{EVI}$ (i.e. RF-RF$_{EVI}$). The same method was used to separate structural and physiological effects of legacy effects on Tr.





### 3.7 Quantifying concurrent and lagged reduction in GPP from drought

Additionally, we compared the estimated legacy effects on GPP to the concurrent drought-induced GPP anomalies. To compute the concurrent reduction in GPP, we summed up all GPP anomalies over each identified drought period. Here,

drought periods were defined as the periods where $WAI_{anom}$ was lower than -1 of standard deviation ($WAI_{SD}$). $WAI_{SD}$ was calculated for each day of year by using a centered 7-day moving window instead of a single value over the whole time series because $WAI_{SD}$ showed a seasonality. This definition only relied on the water availability without considering biospheric responses because WAI directly indicated the water supply for vegetation while GPP could include other factors addition to drought in short periods. We quantified the lagged reduction in GPP at the annual scale as the difference between

$GPP_{anom}$ residuals in legacy years and the median of the model uncertainties. To compare the reduction in GPP across sites, both concurrent and lagged values were normalized relative to averaged total GPP over the growing season.

### 4. Results

### 4.1 GPP time series in drought and legacy years



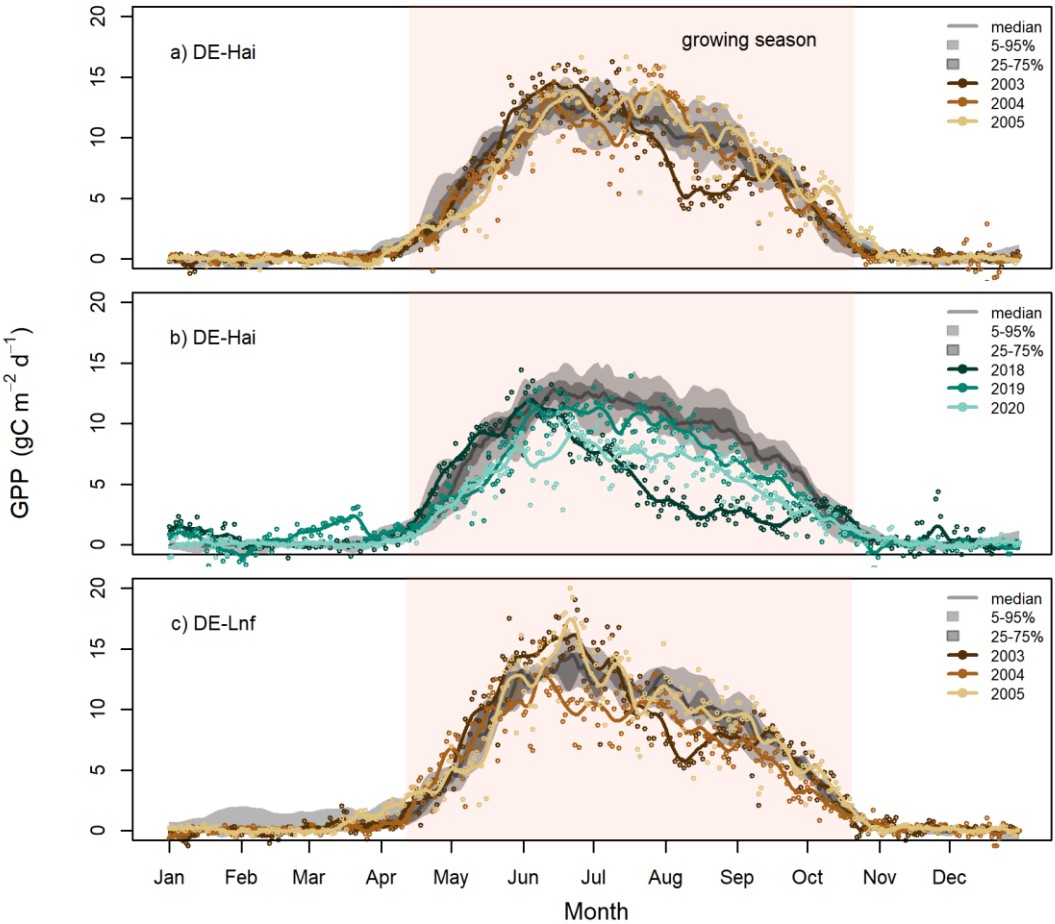

**Figure 2. Daily GPP in the selected drought and legacy years at a), b) DE-Hai (showing the 2003 and 2018 droughts, respectively) and c) DE-Lnf (showing the 2003 drought).** Colored points and lines showed original and smoothed (7-days average) GPP, respectively, in drought and legacy years. The grey lines and shaded areas showed the median, 25th-75th (dark grey), and 5th-95th (light grey) percentiles of GPP, respectively, over non-drought and non-legacy years. The shaded coral areas indicate the average growing seasons of DE-Hai and DE-Lnf.

In Fig. 2, we show the measured absolute GPP time series in the selected drought (2003 and 2018) and legacy years (2004, 2005, 2019, and 2020) together with the long-term median, 25th-75th, and 5th-95th percentiles GPP at DE-Hai and DE-Lnf. In the drought year 2003, GPP was significantly lower than the baseline, defined as the 25th percentile GPP, during July-September at DE-Hai and July-August at DE-Lnf, respectively. In the post-drought years 2004 and 2005, there was no systematic decrease in GPP at DE-Hai, while GPP at DE-Lnf was slightly lower than the baseline during June-August of 2004. During the 2018 drought, GPP significantly differed from the baseline during June-September at DE-Hai. After the 2018 drought, we could not find any systematic decrease in GPP in 2019, while GPP was consistently lower than the baseline from mid-May to September of 2020 at DE-Hai.





## 4.2 Drought legacy effects on GPP: seasonal patterns



**Figure 3. Residuals of GPP anomalies at the seasonal scale in legacy years at a) DE-Hai and b) DE-Lnf.** Residuals of GPP anomalies were characterized by observed minus predicted GPP anomalies (GPP$_{anom}$ residuals). The color lines and bands show the median and 5th-95th percentile GPP$_{anom}$ residuals of ensemble model runs (see Section 3.4), respectively. Negative residuals corresponded to more negative or less positive GPP$_{anom}$ than would be expected given the climate in that year, indicating negative legacies of drought, while positive residuals corresponded to less negative or more positive GPP$_{anom}$, indicating beneficial legacies of drought. The model uncertainties (dark and light grey shaded area, respectively) are characterized by the 25th-75th and 5th-95th quantile ranges of GPP$_{anom}$ residuals in non-legacy years. The black line represents the median of GPP$_{anom}$ residuals in non-legacy years. The ticks denote the start of each month.





At the seasonal scale, residuals of GPP anomalies (GPP$_{anom}$ residuals) showed significant departures from model
uncertainties at both sites (Fig. 3). After the 2003 drought at DE-Hai, we found negative residuals below the 25$^{th}$ percentile
of model residuals in non-legacy years (model uncertainties) during the early and late growing season of 2004 (April-July,
September) and May-June of 2005, and below the 5$^{th}$ percentile for short periods, in April and May of 2004 and May of
2005. After June 2005, residuals were mostly within 5-95% of the model residuals. After the 2018 drought at DE-Hai, we
found negative residuals (below 25$^{th}$ percentile of model residuals) during May, June, August, and September of 2019. In
2020, residuals showed a persistent decrease from May to July, and generally stayed well below the 5$^{th}$ and 25$^{th}$ percentile of
model residuals from mid-May until July and September, respectively.

After the 2003 drought at DE-Lnf, we found persistent negative residuals were below the 25$^{th}$ percentile of model residuals
over almost the complete growing season (from May to October) in 2004 and below the 5$^{th}$ percentile of model residuals for
periods in June-September. In 2005, residuals remained mostly within 25$^{th}$-75$^{th}$ percentiles of model residuals.

**4.3 Drought legacy effects on GPP: annual patterns**

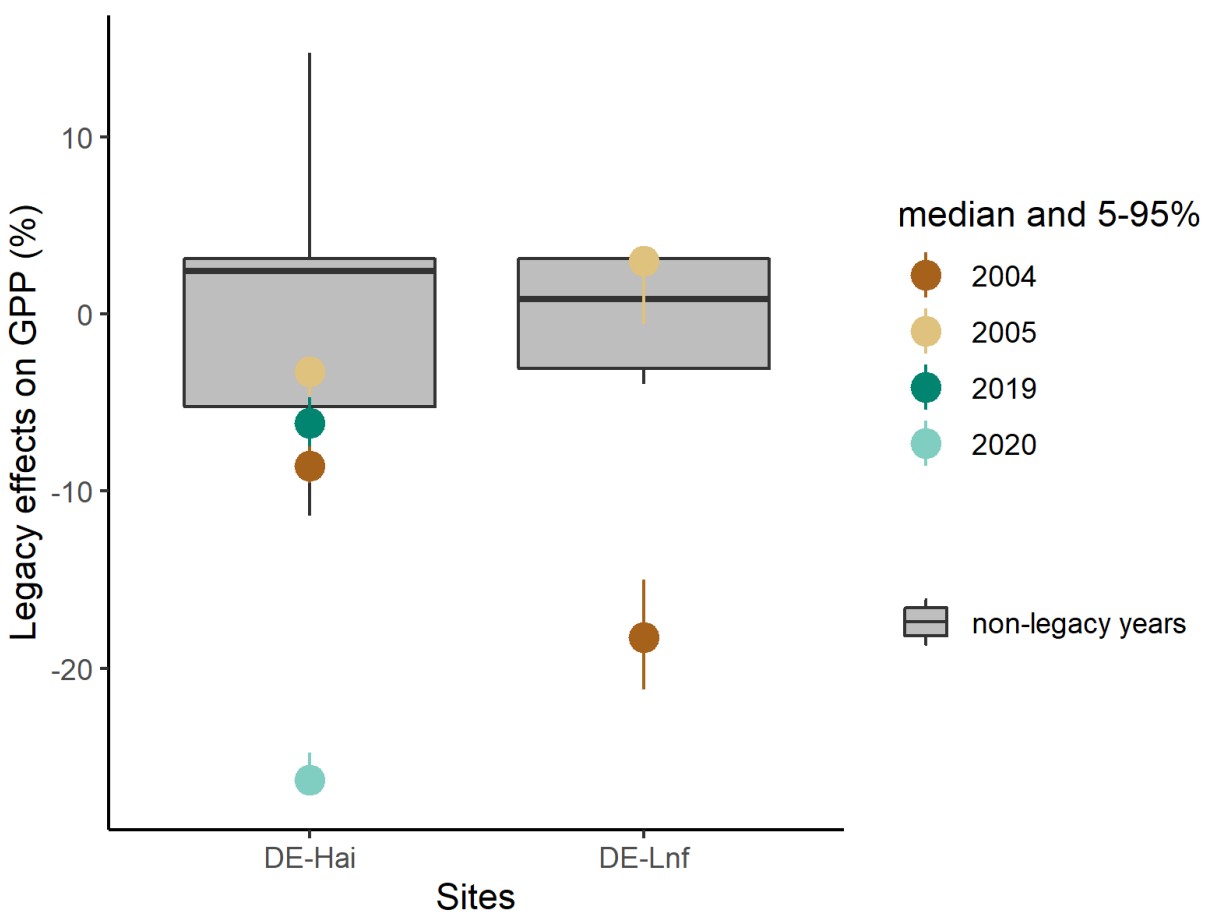



**Figure 4. Integrated residuals of GPP anomalies at the annual scale in legacy years at DE-Hai and DE-Lnf.** The color points and line ranges show the median and 5-95% percentile integrated GPP$_{anom}$ residuals of ensemble model runs (see Section 3.4), respectively. The model uncertainties (the boxplot) are characterized as the 25$^{th}$-75$^{th}$ quantile range of integrated GPP$_{anom}$ residuals in non-legacy years.

There were systematic departures of integrated residuals of GPP anomalies in legacy years from model uncertainties at the annual scale (Fig. 4) although the seasonal patterns varied (Fig. 3). After the 2003 drought at DE-Hai, integrated residuals in 2004 were significantly below the 25$^{th}$ percentile of model residuals, while integrated residuals were within the 25$^{th}$-75$^{th}$ percentiles of model residuals in 2005. After the 2018 drought, integrated residuals in 2019 were near the 25$^{th}$ percentiles of model residuals, while in 2020 integrated residuals were far below the 25$^{th}$ percentile of model residuals.

At DE-Lnf, after the 2003 drought, integrated residuals in 2004 were below the 25th percentile of residuals in non-legacy years, while integrated residuals almost remained within 25$^{th}$-75$^{th}$ percentiles of model residuals in 2005.





## 4.4 Drought legacy effects on GPP due to structural and physiological effects

![Figure 5 plot]

**Figure 5. Residuals of GPP anomalies from RF and RF$_{EVI}$ (see Section 3.6) in legacy years at a) DE-Hai and b) DE-Lnf.** Residuals of GPP anomalies are characterized by observed minus predicted GPP anomalies (GPP$_{anom}$ residuals). The color lines and bands show the median and 5th-95th percentile GPP$_{anom}$ residuals of ensemble model runs (see Section 3.4), respectively. The solid and dashed lines show the residuals based on RF and RF$_{EVI}$, respectively. The model uncertainties from RF$_{EVI}$ (dark and light grey shaded area, respectively) are characterized by the 25th-75th and 5th-95th quantile ranges of GPP$_{anom}$ residuals in non-legacy years. The black dashed line was the median of GPP$_{anom}$ residuals from RF$_{EVI}$ in non-legacy years. The ticks denoted the start of each month.

At the seasonal scale, residuals of GPP anomalies from RF$_{EVI}$ (Res$_{EVI}$) showed significant departures from GPP$_{anom}$ residuals from RF (Res) over some periods at both sites (Fig. 5). At DE-Hai, we found Res$_{EVI}$ was above Res in the early growing season (April-May) of 2004, 2005, 2019, and 2020, and also in the late growing season of 2004 (August-October) and 2019



(August-September). After the 2003 drought, we found negative $Res_{EVI}$ below the 25th percentile of model residuals from
$RF_{EVI}$ in non-legacy years (model uncertainties) during the early and late growing season of 2004 (May-July, September)
and May of 2005, and below the 5th percentile for short periods, in May of 2005. After the 2018 drought, we found negative
$Res_{EVI}$ (below 25th percentile of model residuals) during June of 2019. In 2020, $Res_{EVI}$ showed a persistent decrease from
May to July, and generally stayed well below the 5th and 25th percentile of model residuals from mid-May until July and
September, respectively.

At DE-Lnf, $Res_{EVI}$ was below Res from April to mid-May and significantly above Res almost over the growing season of
2004 (from mid-May to September). We found negative $Res_{EVI}$ below the 25th percentile of model residuals from $RF_{EVI}$ in
non-legacy years (model uncertainties) during June, August, and September of 2004, and below the 5th percentile for short
periods, in June and September of 2004.

**4.5 Drought legacy effects on tree ring width**

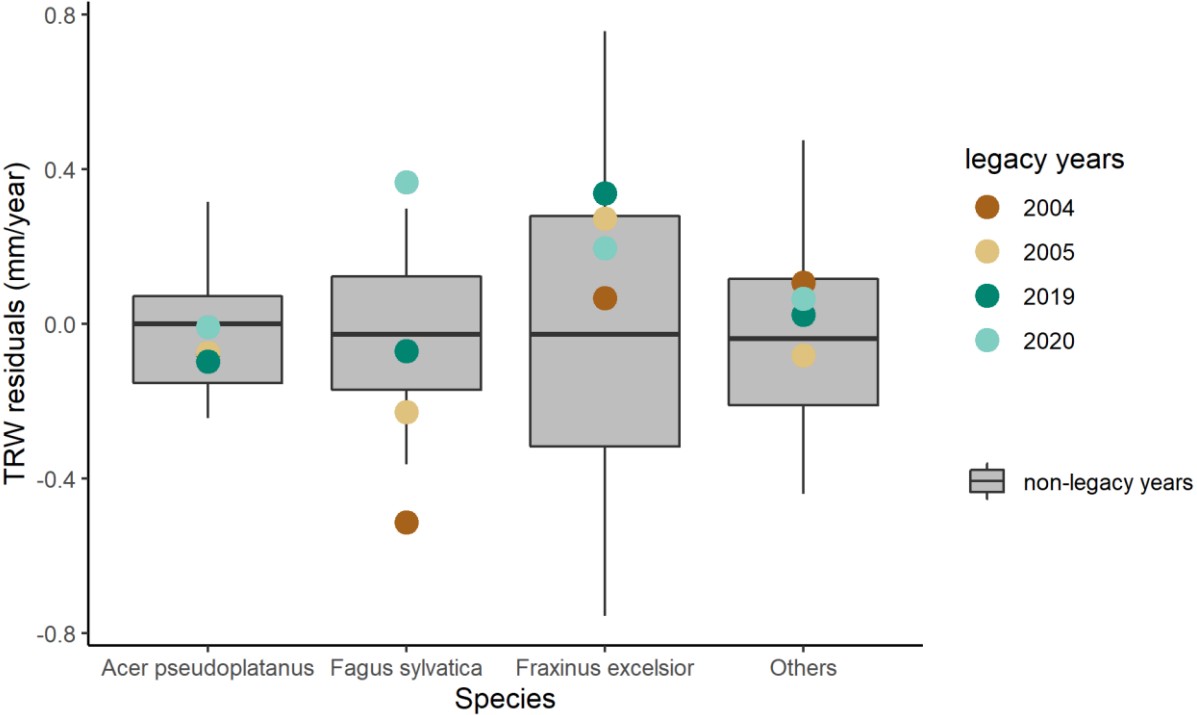

**Figure 6. Residuals of TRW in legacy years at DE-Hai across species.** Residuals of TRW are characterized as observed minus
predicted TRW anomalies (TRW residuals). The model uncertainties (the grey area) are characterized as the 25th-75th quantile range of
TRW residuals in non-legacy years.

To complement the analysis of the legacy effects on GPP at the seasonal and annual scales, we also evaluated legacy effects
on tree growth at the annual scale. In the post-drought years 2004 and 2005, TRW of *Fagus sylvatica* was below the 25th




percentile of model residuals. For species of *Acer pseudoplatanus*, *Fraxnius excelsior*, and others, residuals of TRW were almost within 25th-75th percentiles of model residuals in 2004 and 2005. After the 2018 drought, TRW of *Fagus sylvatica* was within model uncertainties in 2019 while higher than 75th percentile of model residuals in 2020. The residuals of TRW

of *Acer pseudoplatanus*, *Fraxnius excelsior*, and others for 2019 and 2020 were almost within or close to 25th-75th percentiles of model residuals.

## 4.6 Concurrent and lagged reduction in GPP

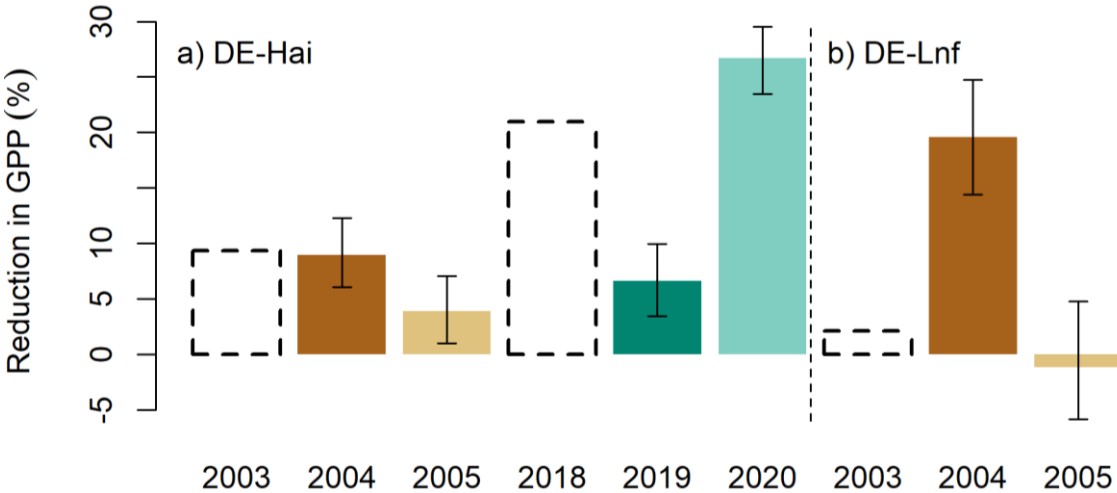

**Figure 7. Concurrent (dashed black bars) and lagged (colored bars) reduction in GPP from the 2003 and 2018 droughts at a) DE-Hai and b) DE-Lnf.** Concurrent impacts in GPP were quantified as the sum of GPP anomalies over drought periods in drought years relative to averaged total GPP over the growing season (see Method). Lagged impacts in GPP are characterized as the difference between GPP$_{anom}$ residuals in legacy years and median of the model uncertainties relative to averaged total GPP over the growing season. Colored bars and error bars show the median and 5-95%, respectively, of lagged reduction in GPP from ensemble model runs.

Finally, we compared the concurrent impacts on GPP with the lagged impacts due to drought. We found that, at DE-Hai, the concurrent reduction in GPP was 9.4% relative to averaged total GPP over the growing season (hereinafter) in 2003, while 6.1-12.3% indirectly reduced in 2004. And in 2018 concurrent reduction in GPP was 21.0%, while 3.5%-10.0% and 23.5-29.6% indirectly reduced in 2019 and 2020, respectively. At DE-Lnf, the concurrent reduction in GPP was negligible in 2003 (2.2%), while we estimated 14.4-24.8% GPP reduction in 2004, which was higher than the corresponding values at

DE-Hai in the same year.





## 5. Discussion

### 5.1 A novel methodology to detect drought legacy effects on GPP

There is limited research on discovering legacy effects of drought on the ecosystem carbon cycle using eddy-covariance
observations (Kannenberg et al., 2019). Here, we propose a residual-based methodology using a random-forest regression
model to detect legacy effects on GPP, and found significant legacy effects on GPP using eddy-covariance data at two
forests in central Germany. There are three advantages to our methodology: 1) capturing the temporal dynamics of legacy
effects at the seasonal scales; 2) separating the influence of meteorological conditions during the post-drought period on
recovery rates; 3) estimating model uncertainties to avoid misinterpreting small residuals as 'legacy effects'.

First, because we used measurements with a high temporal resolution (daily), legacy effects could be determined across
different time scales. Previous studies based on tree-ring or satellite-greenness data have mainly focused on legacy effects at
the annual scale (Anderegg et al., 2015; Wu et al., 2018) or monthly scale (Bastos et al., 2021), but the legacies can be more
ephemeral, for example, if they are expressed only in critical periods of the growing season, as we have found here. Such
temporally confined effects may not necessarily manifest themselves at the annual scale. For example, after the 2003
drought, the annual GPP at DE-Hai in 2005 was close to normal, which was the 25$^{th}$ percentile of model residuals here, but
we found short legacies at the seasonal scale (Fig. 3).

Second, recovery is usually considered when the target variable (i.e. GPP, tree-ring width…) returns to the baseline, usually
based on pre-drought values of the target variable (Bose et al., 2020; González de Andrés et al., 2021; Zhang et al., 2021).
However, meteorological conditions during the recovery period will modulate recovery rates, so that recovery can be
delayed e.g. if a drought is followed by other unfavourable climatic conditions. Hence, the evaluation of possible legacy
effects should be based on the functional relations between the target variable and meteorological conditions. Our model
takes this into account by considering that ecosystems recovered when observed GPP reaches the potential GPP given the
meteorological conditions, rather than the absolute flux.

Finally, our approach allows estimating the uncertainties in estimated legacy effects. Previous studies (Anderegg et al., 2015;
Huang et al., 2018) quantified legacy effects as the residuals between observed and predicted target variables (i.e. tree-ring
width, vegetation indices, …) in legacy years, but were not able to consider uncertainties of their trained models. Yet, it is
crucial to understand if the residuals are caused by model uncertainties or can be interpreted as legacy effects. In this study,
legacy effects are identified only when the model residuals are outside the range of the model uncertainties, so that we are
confident that the legacies reported here are significant and avoid interpreting residuals caused by model error as legacy
effects. A limitation of our approach is that we have to assume that there are no legacy effects in the climate system because
this would potentially bias the interpretation of the residuals.

The methodology we proposed is able to detect the legacy effects of drought on GPP and can be easily applied to other eddy-
covariance sites and variables (i.e. evapotranspiration, transpiration, …), in order to improve our understanding of drought
legacy effects on the ecosystem carbon cycle at different time-scales.




## 5.2 Seasonal and annual legacy drought impacts on GPP

We found that residuals of GPP anomalies (GPP$_{anom}$ residuals) in legacy years were significantly larger than model uncertainties at both seasonal and annual scales at both sites, which indicated strong legacy effects of drought on GPP at least in the two years following the drought events.

We found negative legacies on GPP in the early growing season of all legacy years (2004, 2005, 2019, and 2020) at DE-Hai. Reduced and delayed leaf development due to physiological effects of the 2003 and 2018 droughts (e.g. metabolic damage, non-structural carbohydrates depletion) could result in reduced ecosystem-level photosynthesis (Migliavacca et al., 2009; Rocha and Goulden, 2010; Kannenberg et al., 2019), and could potentially explain negative legacies on GPP at the start of the growing season. In line with this hypothesis, we found the enhanced vegetation index (EVI, a proxy of leaf area index, Fig. S2 and Fig. S3)

at the sites showed lower values than other years in the early growing seasons of 2004, 2005, and 2019 and this delayed spring phenology propagated over the year of 2004 and 2019 with a shift of seasonality. We found consistently lower values of NPP allocated to foliage growth in 2004 than other years (Fig. S4). Furthermore, the detected negative legacies in the early growing season became smaller when adding EVI anomalies as an additional predictor in the random forest model (Fig. 5), indicating that the reduced and delayed leaf development partly explained the estimated legacy

effects by the RF model trained with climate predictors only.

Another possible mechanism explaining legacy effects could be hydraulic damage induced by drought (Anderegg et al., 2013), and therefore insufficient ability of water transport limiting sink strength (Körner, 2015) and photosynthetic capacity (Chen et al., 2010), at least until damage is repaired. If this was the case, transpiration fluxes should be reduced. However, we did not find similar negative legacy patterns on transpiration in the early growing season (Fig. S5a). Therefore, hydraulic

damage did not seem a likely cause of drought legacies on GPP for these events. Overall, we cannot pinpoint the physiological causes of the detected legacy effects due to limited availability of measurements. This calls for establishing more plant-physiological measurements complementing eddy-covariance and TRW measurements to capture sufficient information about plant water relations such as sap flow (Poyatos et al., 2021) and tree water deficit (Nehemy et al., 2021) as well as carbon allocation (Hartmann et al, 2020) to provide a more detailed process understanding of the mechanisms

underlying drought legacy effects.

Negative legacies on GPP in terms of lagged reduction in GPP in 2004 at DE-Lnf (14.4-24.8%) were stronger than those at DE-Hai (6.1-12.3%) in the seasonal and annual scales. The persistence of negative legacies throughout the full growing season in 2004 indicates that the 2003 drought likely caused stronger damage, especially reduced leaf development which was supported by largely reduced negative legacies of RF$_{EVI}$ with EVI comparing to RF without EVI (Fig. 5), on the

ecosystem at DE-Lnf than that at DE-Hai. From the community-level perspective, the stronger legacy effects found at DE-Lnf compared to DE-Hai may have been partly related to differences in forest composition between the two sites (Tamrakar et al. 2018, Pardos et al., 2021). Measurements of GPP at tree species level were not available, therefore we relied on the





legacies found for TRW (reflecting growth), available for individual trees at DE-Hai. It should be noted, though, that the relationship between GPP and growth is complex (Fatichi et al., 2014). Negative legacy effects on TRW of *Fagus sylvatica*,

dominating at DE-Hai, in 2004 and 2005, were found, whereas other co-dominating species (*Acer pseudoplatanus* and *Fraxinus excelsior*) did not show negative legacies. Therefore, the lower resilience of *Fagus sylvatica* compared to other species may have partly resulted in stronger negative legacies at the pure European beech forest at DE-Lnf than at DE-Hai. In addition, contrasting legacy effects of these two sites could also be associated with different age classes and the absolute stand age since the effects of stand age on determining the heat and drought impact on carbon exchange (Arain et al., 2022)

and ecosystem-level photosynthetic capacity (Musavi et al., 2017) have been recognized. However, the evidence of species diversity and age structure effects on legacy effects needs to be further explored using more sites in the future.

Stronger negative legacy effects on GPP in 2020 than those in other legacy years were found at DE-Hai in the seasonal and annual scales. This might be associated with significant tree mortality in the period 2018-2020 (about 6% year$^{-1}$ between 2017 and 2020 compared to less than 1% year$^{-1}$ between 2005 and 2017) mainly caused by the storm *Friedrike* in January

2018 and the heat and/or drought in summer 2018 and 2019 (unpublished data). TRW of *Fagus sylvatica* in 2020, on the contrary, showed positive legacy effects in growth, since only living trees were sampled. This might be explained by the favorable weather conditions in winter/spring 2019/2020 associated with high mineralization rates and reduced competition for nutrients, light and water of the surviving trees (Grossiord, 2020). The TRW data reflected mean growth signals from individual survived trees, while the GPP data reflected mean carbon assimilation at stand level, including positive, negative

or absent legacy effects at individual tree level as well as the reduction of assimilating individuals due to higher tree mortality.

Overall, we found that the lagged impacts of drought on GPP are significant compared with concurrent drought impacts at the two sites studied here. The lagged reduction in GPP resulting from drought is usually not quantified (Ciais et al., 2005; Reichstein et al., 2007), perhaps because separating legacy effects on ecosystem carbon fluxes from observations is

challenging (Kannenberg et al., 2019) and process-based models have been shown to miss such legacy effects (Bastos et al., 2021). This implies that the impact of droughts on ecosystem carbon cycling in most studies might be underestimated.

### 5.3 Importance of deep root-zone soil moisture data

Deep root-zone soil moisture has been recognized as an important water source for vegetation, especially during droughts

(Miguez-Macho and Fan, 2021; Werner et al., 2021). Although soil moisture measurements across three soil layers are available at both sites, the deepest depth (ca. 30cm) cannot capture the entire soil water reservoir available for European beech which has been observed to have non-negligible amounts of fine roots below 30cm across different sites (Leuschner et al., 2004, Gessler et al., 2021).

We tested an initial model using anomalies of soil moisture at three layers as predictors (RF$_{SM}$), and found strong positive

legacy effects in the late growing season in 2019 at DE-Hai (Fig. S6), which however could not be reproduced by any of the





models using soil moisture information from deeper layers (Fig. S6) including the local water balance (WAI, CWD) and the reanalysis data (ERA5). Comparing the predicted time series of $GPP_{anom}$ of the $RF_{SM}$ model with observations, we found the predicted $GPP_{anom}$ became much more negative in the late growing season while observed $GPP_{anom}$ were close to zero (Fig. S7). Therefore, although soil moisture anomalies in the third layer (30cm) were largely negative when the positive residuals

appeared (Fig. S8), soil moisture from layers deeper than 30 cm may maintain the water supply for photosynthesis. Also, we found the evapotranspiration from the shallow layers (0~30cm) estimated by soil moisture decrease was less than the observed evapotranspiration during dry-down periods (Fig. S9), which indicated plant water uptake from layers deeper than 30 cm during dry-down periods, in line with our hypothesis. In summary, these positive patterns are likely due to model errors from incomplete information on the soil-moisture profile rather than actual positive legacy effects.

These results highlight the importance of soil moisture measurements that capture the entire root zone for more reliable understanding of ecosystem functioning, particularly in the case of drought legacy effects.

## 6. Conclusions

The frequency, intensity, duration, and spatial extent of droughts are expected to increase in the next decades due to anthropogenically caused global warming in many regions (IPCC, 2022). Drought not only impacts ecosystems

concurrently, but also can have legacy effects on ecosystem carbon fluxes. We developed a residual-based approach using a random forest regression model to detect drought legacies on gross primary productivity (GPP) using eddy-covariance data. The methodology proposed here allows quantifying significant drought legacy effects on GPP at the sub-seasonal and annual scales. To the best of our knowledge, this is the first time that drought legacies on ecosystem carbon fluxes in observation are quantified using eddy-covariance data. The GPP reduction due to drought legacy effects is of comparable magnitude to

the concurrent drought effects at the studied sites, which confirms the importance of legacy effects. We found contrasting legacy effects at two neighbouring forests with different species and age structures, yet the importance of these factors could not be evaluated. Future studies across a larger range of sites will be needed to understand whether the crucial role of legacy effects is general and on which mediating factors they depend.

**Acknowledgements**

Xin Yu and Markus Reichstein acknowledge funding from the European Research Council (ERC) Synergy Grant "Understanding and Modelling the Earth System with Machine Learning (USMILE)" under the Horizon 2020 research and innovation programme (Grant agreement No. 855187). Xin Yu acknowledges support from the International Max Planck Research School for Global Biogeochemical Cycles. René Orth acknowledges support by the German Research Foundation

(Emmy Noether grant 391059971). Franziska Koebsch and Alexander Knohl acknowledge support by Niedersächsisches

Vorab (ZN 3679), Ministry of Lower-Saxony for Science and Culture (MWK). Martina Mund acknowledges support by the Integrated project CarboEurope-IP (European Commission, Directorate-General Research, Sixth Framework Programme, Priority 1.1.6.3: Global Change and Ecosystem, Contract no. GOCECT-2003-505572), the Max Planck Institute for Biogeochemistry, Germany, the German Research Foundation (DFG) (INST 186/1118-1 FUGG), the German Federal
Ministry of Education and Research (BMBF; research infrastructure ICOS) and the Georg-August-University Göttingen, Germany. Sophia Walther acknowledges funding within the ESA Living Planet Fellowship on the project Vad3e mecum. Benjamin D. Stocker was funded by the Swiss National Science Foundation grant no. PCEFP2_181115. We thank the Warm Winter 2020 initiative of the Integrated Carbon Observation System (ICOS), the administration of the Hainich National Park for the opportunity for research within the National Park, and Martin Jung of Max Planck Institute for Biogeochemistry for
the inspiring discussions.

**Author contributions**

The study was conceived by X. Yu, A. Bastos, R. Orth, M. Reichstein, and M. Bahn. X. Yu implemented the method and performed the data analyses. A. Knohl, A. Klosterhalfen, and F. Koebsch provided the eddy-covariance data. M. Mund
provided the data of tree ring width and net primary productivity of fruits and leaves. J. A. Nelson helped X. Yu to process the transpiration estimation. S. Walther provided and processed the Enhanced Vegetation Index data. B. D. Stocker suggested quantitatively separating structural and physiological effects. M. Migliavacca helped to interpret the results. X. Yu, A. Bastos, R. Orth, M. Reichstein, and M. Bahn prepared the first draft and all authors contributed to discussion of results and the revisions of the manuscript.


**Competing interests**

At least one of the (co-)authors is a member of the editorial board of Biogeosciences. The peer-review process should be guided by an independent editor, and the authors also have no other competing interests to declare.

**Code and data availability**

Eddy-covariance and enhanced vegetation index data used are freely accessible. Tree ring width and net primary productivity of fruits and leaves data are available on request to Martina Mund. Our code is available on request.



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
