# Peer review of "Contrasting drought legacy effects on gross primary productivity in a"

_Biogeosciences, 2022_

## Author Comment (AC1)

**Contrasting drought legacy effects on gross primary productivity in a mixed versus pure beech forest**

Xin Yu, René Orth, Markus Reichstein, Michael Bahn, Anne Klosterhalfen, Alexander Knohl, Franziska Koebsch, Mirco Migliavacca, Martina Mund, Jacob A. Nelson, Benjamin D. Stocker, Sophia Walther, and Ana Bastos. *Biogeosciences Discussion*

Response to Reviewer #1

**Major comments:**

**R1C1: In this manuscript, Yu et al. investigate drought legacy effects in GPP at two contrasting forest types in Germany. This manuscript represents several notable advances, including: 1) direct observation of GPP legacy effects, 2) a method to quantify sub-annual legacies, 3) incorporating uncertainty in legacy effect calculations, and 4) a neat idea to get at the mechanism behind GPP legacies. In addition, the manuscript is written very clearly and is quite compelling to read. This is a great contribution to the literature and only have a few suggestions.**

We thank the reviewer for the overall positive evaluation of our study and for the constructive comments. Below, we provide a point-by-point reply to the reviewer comments.

**R1C2: The approach to calculate a "tree ring width" based on dendrometer bands is interesting. However, due to bark shrinkage and expansion, these processes aren't exactly analogous. I think there needs to be an acknowledgement of this and a discussion of how these biases might play out.**

Thanks for pointing this out. The method we used has - compared to tree cores where TRW is directly measured - three central advantages: permanent, long-term and non-invasive/non-destructive measurement of the same trees, measurement of fresh wood increment, and measurement of the entire stem at breast height (not only of a very small part). The latter two advantages are important for an upscaling to total wood growth at stand level. The main disadvantage of our method - the inclusion of shrinkage and swelling of the bark - is an "accepted" uncertainty. This uncertainty can be accepted  (1) because we used only the annual increment (not as in Mund et al 2010 where monthly growth rates were investigated), (2) because the dominant species is beech that has only a thin bark, (3) because we recorded the final stem diameter of each year in winter, when the water status of the xylem and the bark is relatively constant, and when stem wood or the bark are not affected by frost or late/early growth or water uptake, (4) because in this study we were interested only in the interannual variability of stem growth, which is less affected by shrinkage and swelling at the described temporal scale than absolute growth rates. Thus, in our case the bark-associated uncertainty results mainly from annual bark increment that is small compared to stem growth in the studied species (no pine or oak). Nevertheless, to be clear and to avoid any misunderstanding, we have changed the term 'tree ring width' to 'radial increment', and have added the description in line 106 of section 2.3:

*'Annual radial increment (RI) was calculated from permanent band dendrometers which measures change in stem girth (or circumference) over bark. The effect due to the inclusion of shrinkage and swelling of the bark is a negligible uncertainty for four reasons: 1) we used only the annual increment, 2) the dominant species is beech that has only a thin bark, 3) we recorded the final stem diameter of each year in winter, when the water status of the xylem and the bark is relatively constant, and when stem wood or the bark are not affected by frost or late/early growth or water uptake, and 4) in this study we were interested only in the interannual variability of stem growth,*

*which is less affected by shrinkage and swelling at the described temporal scale than absolute growth rates.'*

**R1C3: How well does the RF model predict GPP during drought years, if trained on non-drought data? Or, just trained on a subset of droughts and used to predict other droughts? The answer to this question has implications for the interpretation of the legacy effect calculation.**

[Figure]

Figure R1.1. Observed (OBS) and predicted (RF) GPP anomalies from trained models using subsets of the training dataset a) at DE-Hai in 2003 and 2018, b) at DE-Lnf in 2003.

We trained RF models using a subset of training dataset (sub) to test the model performance during drought years. For example, in order to test the model performance in 2003, the data in 2003 was excluded from the training dataset and the trained model was used to predict GPP anomalies in 2003 given meteorological variables. The same strategy was applied to 2018 at DE-Hai and 2003 at DE-Lnf. We compared observed (OBS) and predicted (RF) GPP anomalies from subset-trained RF models a) in 2003 and 2018 at DE-Hai and b) in 2003 at DE-Lnf. We found that RF matches OBS GPP relatively well at DE-Hai during the growing seasons of 2003 and 2018 (R2 of 0.49 and 0.35, respectively) and captures the GPP decrease during the drought period of 2003, but estimates a weaker decrease in the 2018 drought. This is thanks to the fact that the 2018 drought is more extreme than the 2003 drought, and the RF model of which training dataset including the 2018 drought is expected to capture the 2003 drought which is within the range of training data. However, the prediction skills of the RF model in 2018 at DE-Hai and 2003 at DE-Lnf are limited, because of the limited range of training data. Therefore, in the manuscript, to avoid extrapolation

beyond the range of training data, we included drought and non-drought years in our training dataset, which maximizes the range of training data. The meteorological conditions in legacy years are within the range of training data, and the trained model is expected to be able to predict reasonable potential GPP anomalies in legacy years. Furthermore, such RF prediction inaccuracies are included in our uncertainty calculation, and the respective uncertainty ranges have been shown in the main figures.

**R1C4: Along those lines, there also needs to be some information regarding model fit, predictive ability, variable importance, etc. in the methods or results. Does model fit differ across sites, years, etc? It seems like a model with a lot of uncertainty at one site or one year may drastically alter the legacy effect calculation. What variables are most important for predicting fluxes at these sites?**

[Figure]

Figure R1.2. Out of bag (OOB) scores of RF models at DE-Hai and DE-Lnf.

We reported here the Out of bag (OOB) scores, which indicate the model prediction ability (closer to 1 is better), of RF models at DE-Hai and DE-Lnf (Figure R1.2). Since using leave-one-out strategy (see Section 3.4), each RF model for a resulting time series has its own OOB score. The medians of all these OOB scores are ~0.7 and ~0.8 at DE-Hai and DE-Lnf, respectively, which are good enough to allow predicting reasonable potential GPP anomalies in legacy years. We agree that the uncertainty from one year could alter the legacy effects calculation, but the difference is not strong enough to change the main legacy signals. This can be seen in the different quantiles of legacy effects in Figure 3 based on the ensemble runs (see Section 4.2), and the band range is relatively narrow.

[Figure]

Figure R1.3. Variable importance, indicated by increased MSE, of RF models at DE-Hai and DE-Lnf.

Here, we also showed the increased MSE, indicating variable importance, from ensemble model runs at a) DE-Hai and b) DE-Lnf (Figure R1.3). The water availability index is the most important explanatory factor at both sites, followed by incoming radiation at DE-Hai and the phenological stage (given by potential radiation) at DE-Lnf.

We have added Figure R1.2 and R1.3 to supplementary materials as Figure S2 and S3, adjusted the order of other figures in the supplement, and added the description in line 179 of section 3.4:

'*First, all daily data in non-legacy years were used as input for the RF model to determine the relationships between anomalies of GPP (GPP$_{anom}$) and anomalies of hydro-meteorological variables (SW_IN$_{anom}$, TA$_{anom}$, VPD$_{anom}$, and WAI$_{anom}$) along with absolute values of SW_IN_POT to capture seasonal variations in the response of ecosystems to hydro-meteorological conditions. These relationships represented long-term controls of climate on GPP, including drought events and near-average or wet conditions. The Out of bag (OOB) of scores indicating the prediction ability of RF models were ~0.7 and ~0.8 (where zero indicates no skill and 1 denotes perfect performance) at DE-Hai and DE-Lnf, respectively (Fig. S2). WAI$_{anom}$ is the most important explanatory factor at both sites, followed by SW_IN$_{anom}$ at DE-Hai and the phenological stage (given by SW_IN_POT) at DE-Lnf (Fig. S3).*'

**R1C5: It doesn't seem like there is any mention in the methods regarding how the length and size of legacies were calculated. It is implied that GPP recovers when it hits the uncertainty boundary, but not explicitly stated.**

We briefly mentioned how the length of legacies was selected in Section 3.3 (Lines 153-158 of the original manuscript).

But following the reviewer's comment, we realize that this statement is not explicit enough, therefore we have rephrased it in line 156 as follows:

'*we selected a legacy period of two years and this choice was justified by the fact that GPP anomalies residuals returned to the range of model uncertainties (i.e. 25$^{th}$-75$^{th}$ percentiles of model residuals), which is considered as the point when GPP recovers, in 2005 (see Section 4.3) following the 2003 drought at both sites and, for 2018 at DE-Hai, data was only available up to 2020.*'

**Minor comments:**

**L118: What constitutes "good" gapfilling?**

It refers to the standard eddy-covariance gap filling algorithm (Pastorello et al., 2020; Reichstein et al., 2005). At the half-hourly (or hourly) scale, there are three different reliable degrees of gap filled data, which are good quality gapfill, medium, and poor. The reliability of gap fill data depends on the availability of data and the size of the time window. For more details, please check Appendix A in Reichstein et al., 2005.

**L133: I might be missing something, but this doesn't seem to mention how WAI is calculated. WAIt depends on the calculation of WAIt-1, which is undefined. So, the definition seems circular.**

We have added more details in lines 133-136 of section 3.2 about WAI calculation.

*'Therefore, we used a bucket model approach based on observed evapotranspiration and precipitation to estimate a vegetation water availability index, WAI (Tramontana et al., 2016), calculated as:*

$$WAI_0 = WAI_{wam\text{-}up} \qquad\qquad (1)$$
$$WAI_t = min(WAI_{max}, WAI_{t\text{-}1} + P_t - ET_t) \qquad\qquad (2)$$

*where $WAI_0$ is the initial value of the water availability index (WAI), $WAI_{warm\text{-}up}$ is the end value of WAI from the warm-up of the bucket model (Eq. 1). To warm up the bucket model, we ran it 5 times through the first year before starting the actual computation across all considered years. $WAI_{t\text{-}1}$ (mm) and $WAI_t$ (mm) were WAI at time step t-1 and t, respectively, $P_t$ (mm) and $ET_t$ (mm) were precipitation and evapotranspiration at time step t (Eq. 2). We set the bucket size (i.e. $WAI_{max}$) as the maximum cumulative water deficit (CWD) at each site. The estimated bucket sizes were 205mm and 191mm at DE-Hai and DE-Lnf, respectively.*

*Additionally, we calculated the CWD, which was estimated from cumulative differences between observed evapotranspiration and precipitation over periods where cumulative net water loss from the soil ($\Sigma$ (ET-P)) is positive.'*

**L197-199: Detrended how?**

We have added one sentence to describe how to detrend in line 199 of section 3.5.

*'We detrended the time series of all variables by removing any significant long-term linear trend detected using the Mann-Kendall test (Kendall, 1948).'*

**L198: Is this annual, or a mean from the growing season? The latter would probably be more relevant.**

[Figure]

Figure R1.4. Residuals of radial increment (RI) in legacy years at DE-Hai across species.

We agree and have changed to growing-season mean values, but still, only negative legacy effects in 2004 were found on RI of *Fagus sylvatica*. We found only slightly positive or no legacy effects on RI of *Fagus sylvatica* in 2020, which is different from the original results using annual values where RI of *Fagus sylvatica* in 2020 showed strong positive legacy effects. We have replaced the original Figure 6 in the manuscript as Figure R1.4 and have corrected the relevant description in line 198 of section 3.5:

[revised manuscript text omitted]

---

## Author Comment (AC2)

**Contrasting drought legacy effects on gross primary productivity in a mixed versus pure beech forest**

Xin Yu, René Orth, Markus Reichstein, Michael Bahn, Anne Klosterhalfen, Alexander Knohl, Franziska Koebsch, Mirco Migliavacca, Martina Mund, Jacob A. Nelson, Benjamin D. Stocker, Sophia Walther, and Ana Bastos. *Biogeosciences Discussion*

**Response to Reviewer #2**

**R2C1: The authors present a sophisticated collaborative work and the development of a new method to separate GPP legacy effects. The analysis clearly demonstrates the value and importance of long term flux measurements with eddy covariance in combination with biometric data for the evaluation of concurrent and legacy effects of ecosystem GPP on different temporal scales. The potential applicability to other ecosystems is attractive as well. Only very few remarks needed for clarification.**

We thank the reviewer for the encouraging evaluation of our manuscript and for constructive comments and suggestions. Below, we provide a point-by-point response to the comments.

**R2C2: Regarding the importance of changes in the energy balance caused by drought and legacy effects, a bit more evaluation of evapotranspiration would improve the paper even more. Even though transpiration seems not to be influenced by drought / drought legacy here, it is unclear whether the term 'transpiration' in the manuscript is standing for 'evapotranspiration' from eddy covariance data.**

[Figure]

Figure R2.1. Residuals of evapotranspiration anomalies at the seasonal scale in legacy years at a) DE-Hai and b) DE-Lnf.

In Figure R2.1, we show the legacy effects on evapotranspiration (ET) calculated using the same methodology we applied for GPP and which is described in Section 3.4. We found the legacy effects on ET are small compared to the legacies on GPP at both sites. Nevertheless, evapotranspiration includes not only transpiration but also soil evaporation and interception evaporation as well. Given a certain amount of energy, even though transpiration decreases due to legacy effects caused by plant hydraulic damage, it could be compensated by increased soil evaporation. In the end, the amount of evapotranspiration might remain unchanged. Therefore, in the manuscript, we quantified legacy effects on transpiration estimated by the TEA (Transpiration estimation algorithm) approach (Nelson et al., 2018) to infer possible legacy effects due to plant hydraulic damage, which was briefly described in line 194 of section 3.4.

'The same method was used to quantify legacy effects on transpiration (Tr)'

After the reviewer's comment, we have clarified the respective paragraph in line 194 of the revised manuscript.

*'In order to infer possible legacy effects due to plant hydraulic damage, the same method was used to quantify legacy effects on transpiration (Tr) estimated by the TEA (Transpiration estimation algorithm) approach (Nelson et al., 2018). The TEA approach first isolates the periods when evapotranspiration is most likely dominated by transpiration. Then, a quantile random forest model (Breiman, 2001; Meinshausen and Ridgeway 2006) is trained during the separated periods and transpiration can be estimated at every time step. More detail can be found in Nelson et al., 2018. Not using evapotranspiration (ET) is because given a certain amount of energy even though Tr decreases due to plant hydraulic damage but it could be compensated by increased soil evaporation, and the amount of ET might remain unchanged.'*

**R2C3: Mortality of trees is mentioned to be already caused by droughts. Can the effect of mortality / less trees over time be separated already? Have these trees been in the flux footprint? It should also be mentioned whether the biomass data from dendrometers and the litter harvest were from within the footprint.**

Unfortunately, the methodology and available data do not allow separating the mortality effects, but we can confirm that there is significant tree mortality in the period 2018-2020 in the flux tower footprint. This is based on observations by the site PIs, who co-authors this study. We also confirm that dendrometers, leaves, and fruits data were collected within the main footprint area. We have added this information in line 106 of section 2.3:

*'Annual mean tree ring width (TRW) was measured via permanent band dendrometers. The dendrometer trees represented the main species and their respective size classes of the main footprint at DE-Hai for the years 2003 to 2020. Because of technical constraints, damages and a natural dieback of single trees, the number of measurement trees per year varied between 54 and 95. Net primary productivity (NPP) of fruits for the years 2003 to 2020, and NPP of leaves for the years 2003 to 2016 resulted from litter samplings (25-29 traps) within the main footprint area of the flux tower'*

And in line 403 of section 5.2:

*'This might be associated with significant tree mortality in the forest covering the main footprint of the flux tower in the period 2018-2020 (about 6% year$^{-1}$ between 2017 and 2020 compared to less than 1% year$^{-1}$ between 2005 and 2017) mainly caused by the storm Friedrike in January 2018 and the heat and/or drought in summer 2018 and 2019 (unpublished data)'*

**Specific remarks:**

**L 212ff: could you clarify a bit more the description of the model setting with EVI anomalies? It seems not to be totally clear how structural effects are removed**

We have added more details and rephrased the description in line 213 of section 3.6:

*'Combining GPP and satellite-based EVI allows separating these structural and physiological effects. To do this separation, we used two model settings: 1) RF, which was the original setting described in section 3.4, included both structural and physiological effects; 2) RF$_{EVI}$, which added EVI anomalies as an additional predictor to the original model, only included physiological effects, because structural effects have been reflected by the predictor EVI anomalies and GPP$_{anom}$ residuals from this model are expected to be caused by physiological effects. Therefore, physiological legacy effects on GPP were quantified as GPP$_{anom}$ residuals from RF$_{EVI}$ while structural legacies were quantified as the difference between GPP$_{anom}$ residuals from RF and RF$_{EVI}$ (i.e. RF-RF$_{EVI}$).'*

**L 223: "…other factors in addition to…"**

corrected.

**L 331/332: "…using eddy-covariance data at two forests in central Germany in the same climate but with different management and species composition." I suggest to repeat here briefly what these forests have in common and where they differ.**

We have added the information.

**L 338: "…if they appear only in critical periods of the growing season,…" –check formulation**

corrected.

**L 349: "Finally, our approach allows determining the uncertainties in estimated legacy effects…" replace one 'estimate'**

corrected.

**L 365: "…negative legacies on GPP (reduced uptake) in the…" -just for the reader's convenience**

We have added it.

**L 399: "…of stand age the heat and drought impact on carbon…."**

We have rephrased it.

**L 431 + 432: this should probably be evaporation instead of evapotranspiration**

Thanks, this should be evapotranspiration, because this refers to the total evapotranspiration including soil evaporation and transpiration from the shallow layers (0~30 cm).

**Fig. 2:**

**As a) represents DE-Hai for 2003 and following years and b) represents DE-Hai for 2018, I suggest to write such:**

**Figure 2: "Daily GPP in the selected drought and legacy years at a), DE-Hai 2003, b) DE-Hai 2018 and c) DE-Lnf showing the 2003 droughts and following legacy years, respectively."**

We have rephrased it.

**Similar for Fig. S2**

We have rephrased it.

**Fig 5: seasonal GPP anomalies: lines ResEVI (structural effect) in figures hard to distinguish from Res. Could you e.g. zoom in to the periods discussed?**

We have zoomed in on the periods discussed in Figure R2.2 but have found it is too informative. Therefore we have added Figure R2.2 to the supplementary materials and have added the description in line 285 of Figure 5.

[Figure]

Figure R2.2. Residuals of GPP anomalies from RF and $RF_{EVI}$ (see Section 3.6) in legacy years at a) DE-Hai and b) DE-Lnf.

'*Figure 5. Residuals of GPP anomalies from RF and $RF_{EVI}$ (see Section 3.6) in legacy years at a) DE-Hai and b) DE-Lnf.* Residuals of GPP anomalies are characterized by observed minus predicted GPP anomalies ($GPP_{anom}$ residuals). The color lines and bands show the median and $5^{th}$-$95^{th}$ percentile $GPP_{anom}$ residuals of ensemble model runs (see Section 3.4), respectively. The solid and dashed lines show the residuals based on RF and $RF_{EVI}$, respectively. The model uncertainties from $RF_{EVI}$ (dark and light grey shaded area, respectively) are characterized by the $25^{th}$-$75^{th}$ and $5^{th}$-$95^{th}$ quantile ranges of $GPP_{anom}$ residuals in non-legacy years. The black dashed line was the median of $GPP_{anom}$ residuals from $RF_{EVI}$ in non-legacy years. The ticks denoted the start of each month. Figure S4 shows the results for April-June and August-October at DE-Hai in more detail.'